# Allergen Diversity and Abundance in Different Tissues of the Redclaw Crayfish (*Cherax quadricarinatus*)

**DOI:** 10.3390/foods13020315

**Published:** 2024-01-19

**Authors:** Emily M. Jerry, Shaymaviswanathan Karnaneedi, Thimo Ruethers, Dean R. Jerry, Kelly Condon, Andreas L. Lopata

**Affiliations:** 1Molecular Allergy Research Laboratory, College of Public Health, Medical and Veterinary Sciences, James Cook University, Townsville, QLD 4811, Australia; emily.jerry@my.jcu.edu.au (E.M.J.); shaymaviswanathan.karnaneedi@my.jcu.edu.au (S.K.); thimo.ruethers@jcu.edu.au (T.R.); 2Australian Institute of Tropical Health and Medicine, James Cook University, Townsville, QLD 4811, Australia; 3ARC Research Hub for Supercharging Tropical Aquaculture through Genetic Solutions, James Cook University, Townsville, QLD 4811, Australia; dean.jerry@jcu.edu.au (D.R.J.); kelly.condon@jcu.edu.au (K.C.); 4Centre for Sustainable Tropical Fisheries and Aquaculture, College of Science and Engineering, James Cook University, Townsville, QLD 4811, Australia; 5JCU AquaPATH Detection Laboratory, James Cook University, Townsville, QLD 4814, Australia; 6Centre for Food and Allergy Research, Murdoch Children’s Research Institute, The Royal Children’s Hospital, 50 Flemington Road, Parkville, VIC 3052, Australia; 7Tropical Futures Institute, James Cook University, 149 Sims Drive, Singapore 387380, Singapore

**Keywords:** allergens, food allergy, shellfish, redclaw, tissue variation, heat treatment, cross-reactivity, proteomics, liquid chromatography-mass spectrometry, SDS-PAGE, immunoblot, tropomyosin

## Abstract

Shellfish allergy affects ~2.5% of the global population and is a type I immune response resulting from exposure to crustacean and/or molluscan proteins. The Australian Redclaw crayfish (*Cherax quadricarinatus*) is a freshwater species endemic to and farmed in northern Australia and is becoming an aquaculture species of interest globally. Despite being consumed as food, allergenic proteins from redclaw have not been identified or characterised. In addition, as different body parts are often consumed, it is conceivable that redclaw tissues vary in allergenicity depending on tissue type and function. To better understand food-derived allergenicity, this study characterised allergenic proteins in various redclaw body tissues (the tail, claw, and cephalothorax) and how the stability of allergenic proteins was affected through cooking (raw vs. cooked tissues). The potential of redclaw allergens to cross-react and cause IgE-binding in patients allergic to other shellfish (i.e., shrimp) was also investigated. Raw and cooked extracts were prepared from each body part. SDS-PAGE followed by immunoblotting was performed to determine allergen-specific antibody reactivity to sarcoplasmic calcium-binding protein and hemocyanin, as well as to identify redclaw proteins binding to IgE antibodies from individual and pooled sera of shrimp-allergic patients. Liquid chromatography-mass spectrometry (LC/MS) was utilised to identify proteins and to determine the proportion within extracts. Known crustacean allergens were found in all tissues, with a variation in tissue distribution (e.g., higher levels of hemocyanin in the claw and cephalothorax than in the tail). The proportion of some allergens as a percentage of remaining heat-stable proteins increased in cooked tissues. Previously described heat-stable allergens (i.e., hemocyanin and sarcoplasmic calcium-binding protein) were found to be partially heat-labile. Immunoblotting indicated that shrimp-allergic patients cross-react to redclaw allergens. IgE-binding bands, analysed by LC/MS, identified up to 11 known shellfish allergens. The findings of this study provide fundamental knowledge into the diagnostic and therapeutic field of shellfish allergy.

## 1. Introduction

Shellfish allergy affects 0.5–2.5% of the global population and is defined as an immune response resulting from exposure to crustacean or molluscan tissues [1,2]. Allergens in shellfish are typically muscle-related proteins, such as tropomyosin (TM), arginine kinase (AK) and myosin light chain (MLC), in addition to hemolymph-associated allergens, such as hemocyanin (HC) [3,4]. Shellfish allergy has one of the highest rates of food-induced anaphylaxis, affecting 42% of shellfish-allergic adults and 12–20% of children [5]. There is currently no effective treatment for shellfish allergy; thus, management requires strict elimination of the offending foods [6]. However, avoiding the consumption of shellfish may not be enough to avoid allergic reactions, as certain allergens in shellfish have been considered cross-reactive with other arthropods, such as dust mites [7].

In Western food cultures, typically, the tail muscle of crustaceans is consumed, although, in large-clawed crustaceans (e.g., lobsters, crayfish and crabs), the muscle tissue from the claw is also eaten. In eastern cultures (e.g., Asia-Pacific), additional body parts are also consumed (e.g., hepatopancreas and gonads) or used whole in broths and soups [8]. Research into the differential distribution of allergenicity among crustacean tissues has been poorly investigated within the current literature despite cultural differences in shellfish consumption [4,9]. The few studies that have investigated differential tissue allergenicity highlighted that patients may display allergic sensitivity after consuming shrimp cephalothorax tissues but test negative in commercial diagnostic tests using shrimp extracts that are usually based on tail muscle [9,10,11]. For improved clinical diagnosis of shellfish allergy, it is important to understand how different types of tissues differ in diversity and abundance of allergenic proteins. Cooking has also been shown to change the protein composition of tissues by degrading heat-labile proteins. As several allergens are heat-stable, cooking can increase the proportion of heat-stable allergens and thereby increase the allergenicity of tissues consumed [12]. The effect of heat treatment on crustacean allergens in different tissues, however, has not been fully explored.

The Australian Redclaw crayfish (*Cherax quadricarinatus*; termed redclaw hereafter) is a freshwater crayfish endemic to northern Australia. The species has been cultured in Australia since the early 1990s and is now farmed through Southeast Asia, China and South America [13]. Redclaw is a member of the crustacea order Astacidea, in contrast to the more commonly consumed shrimps and crabs (order Dendrobranchiata and Brachyura, respectively). Despite redclaw being commonly eaten as food, no allergens have been officially characterised from the species, and there are limited studies on the cross-sensitisation between redclaw and other edible shellfish. However, redclaw have been demonstrated to elicit IgE-binding and cross-sensitivity with other related crustacea [14]. Cross-sensitisation among crustaceans and between crustaceans and molluscs is often observed, with approximately 80–100% of some crustacean allergens being cross-reactive between other crustacean species, in particular the pan-allergen TM [14]. In contrast, cross-sensitisation between molluscs and crustaceans is often much lower, often less than 20% [15]. As many allergenic proteins in shellfish are functionally important and evolutionary conserved, redclaw allergens may cross-react and elicit IgE binding from patients allergic to other crustacean species.

Due to a lack of knowledge on the allergenicity of this species, this study characterised the redclaw allergome, determined the distribution and proportion of allergens among three different body parts, and evaluated how cooking affected the heat stability of allergens as well as the allergen proportion. Finally, this study examined the potential of redclaw allergens to cross-react with sera from shrimp-allergic patients.

## 2. Materials and Methods

### 2.1. Redclaw Tissue Collection

Thirty adult redclaw were received from the Australian Crayfish Hatchery Pty Ltd., Townsville, QLD, Australia. Redclaws were separated by sex and initially acclimated at ambient room temperature in two flow-through 5000 L tanks for 2 weeks. The redclaw were then further reared for 4 weeks at 28 °C in five 50 L tanks within a freshwater recirculating aquaculture system, whereafter, they were harvested, immediately euthanised and held on ice in preparation for protein extraction.

### 2.2. Preparation of Redclaw Protein Extracts

Raw and cooked (termed heated hereafter) redclaw extracts were prepared according to the method described previously [16]. Tissues from the claw, tail and cephalothorax were minced and homogenised in PBS at 4 mL/g of tissue. To obtain the heated extracts, aliquots of homogenised raw extracts were heated at 90 °C in a water bath for 20 min. The extracts were then incubated at 4 °C, stirred overnight and centrifuged at 20,000× *g* for 20 min at 4 °C. The supernatant was then filtered through a glass fibre filter and finally through a 0.45 µm membrane filter (MicroScience, Taren Point, NSW, Australia).

### 2.3. Quantification of Protein Content

The total protein concentration from both raw and heated extracts was determined and standardised to 1 μg/μL using a Pierce™ bicinchoninic acid assay (BCA) kit (Pierce Biotechnology Inc., Rockford, IL, USA) as per the manufacturer’s instruction. Pre-diluted bovine serum albumin (BSA; Pierce Biotechnology Inc.) was used as the protein standard.

### 2.4. SDS-PAGE

Extracts from 10 redclaw specimens were resolved using a reducing sodium dodecyl sulphate–polyacrylamide gel electrophoresis (SDS-PAGE) method [17] alongside a Precision Plus Protein Dual Color Standards molecular weight marker (BioRad, Hercules, CA, USA). SDS-PAGE gels were stained with Coomassie Brilliant Blue [18] and visualised using the Odyssey^®^ CLx Imaging System (LI-COR Biosciences, Lincoln, NE, USA). The separation of proteins by SDS-PAGE was then compared between 10 individual body parts for subsequent pooling. Aliquots of 100 µL per sample were aliquoted, pooled (total ~1 mL) and utilised for the generation of extracts from raw and heated body parts.

### 2.5. Immunoblotting with Allergen-Specific Antibodies

Proteins separated by SDS-PAGE were transferred to nitrocellulose membranes using a Trans-Blot^®^ SD Semi-Dry Electrophoretic Transfer Cell (BioRad, Hercules, CA, USA). Membranes were blocked with 1× casein and washed, then incubated with in-house generated monoclonal mouse anti-shrimp sarcoplasmic calcium-binding protein (SCP) (1:1000 dilution) or by using in-house generated polyclonal goat anti-shrimp HC (1:50,000 dilution) primary antibodies. After washing, the membranes were incubated with anti-mouse antibody conjugate with LI-COR IRDye 800CW (1:10,000 dilution) (LI-COR Biosciences, Lincoln, NE, USA), or anti-goat IgG antibody conjugate with Thermo Dylight^®^ 800 4X PEG (1:10,000 dilution) (Thermofisher Scientific, Waltham, MA, USA). The membranes were then visualised using the Odyssey^®^ CLx Imaging System (LI-COR Biosciences).

### 2.6. Liquid Chromatography-Mass Spectrometry (LC/MS)

Protein extracts were analysed by liquid chromatography-mass spectrometry (LC/MS) for the presence and proportion of 11 World Health Organisation and International Union of Immunological Societies (WHO/IUIS) registered crustacean allergens and five potential allergens (PAs) (evidence of allergenicity, but not officially registered) (Table 1). In-solution tryptic digest was performed using an S-Trap™ (ProtiFi, Fairport, New York, NY, USA) method following the manufacturer’s instructions. Eluted peptides were analysed by LC/MS on an LTQ Orbitrap Elite (Thermo Scientific) with a nanoESI interface in conjunction with an Ultimate 3000 RSLC nanoHPLC (Dionex Ultimate 3000) (Thermofisher Scientific, Waltham, MA, USA). Label-free quantification was conducted for the proteins in each extract using MaxQuant v2.2 [19] against the in-house database of decapod suborder Pleocyemata, downloaded from the National Center for Biotechnology Information (NCBI). The search parameters were as follows: the specific digestion enzyme was trypsin/P, with the maximum missed cleavages being 2. The variable modifications included oxidation (M) and deamidation (NQ). The fixed variables included carbamidomethyl (C). The allergenic and potentially allergenic proteins that were searched for are presented in Table 1. Known allergens were defined as those registered in the WHO/IUIS and have been repeatedly shown to have clinical reactivity in shellfish-allergic patients. PAs were defined as proteins that have some evidence of IgE-binding in shellfish-allergic patients but are not registered in the WHO/IUIS.

### 2.7. Patient IgE-Antibody Immunoblotting

Shrimp-allergic patient sera (ethics approval was obtained from the Ethics Committee of James Cook University, Approval Numbers: H4313 and H6829; and written informed consent was obtained from all patients.) were selected with three selection criteria, including a clinical confirmation of shrimp allergy, positive shrimp skin prick test (SPT), and a positive sIgE test on a MADx ALEX^2^™ chip to shrimp mix (f24) (Macro Array Diagnostics, Lemböckgasse, Vienna, Austria). A total of 11 adults and 13 paediatric patient sera were chosen based on these criteria (Appendix A).

Raw and heated claw, tail and cephalothorax protein extracts were used to determine the cross-reactivity with IgE antibodies in sera from shrimp-allergic patients. SDS-PAGE was performed using raw and heated tail extracts, transferred to a nitrocellulose membrane, and blocked. The membranes were then incubated with patient sera using a Surfblot method (Idea Scientific Company, Minneapolis, MN, USA) [20]. The membranes were incubated with mouse anti-human IgE antibody (1:1000 dilution) (Santa Cruz Biotechnology, Inc., Dallas, TX, USA), followed by goat anti-mouse IgG antibody conjugate with Li-Cor IRDye 800CW (LI-COR Biosciences). IgE antibody binding and intensity were visualised using an in-gel tryptic digestion [18].

### 2.8. Identification of IgE-Binding Proteins by Liquid Chromatography-Mass Spectrometry (LC/MS)

For the identification of IgE-binding proteins by LC/MS, six sera with the most representative IgE-binding patterns were pooled. SDS-PAGE, membrane transfer and blocking were performed. Representative IgE-binding bands were excised from the SDS-PAGE, and tryptic digestion was performed as described earlier.

### 2.9. Amino Acid Sequence Analysis

Amino acid sequences for TM, AK, HC, MLC, SCP and troponin C were obtained from the protein NCBI database for various crustacean species. A multiple sequence alignment was performed using Clustal Omega [21,22] under the default parameters using an input order. Clustal Omega analysis provided alignments of all crustacean sequences for each allergen, as well as sequence similarity. The corresponding alignments were exported and then imported and viewed in Jalview v2 [23]. In Jalview, the sequence alignments were wrapped and annotated. The Jalview file was then imported into MEGA v.11 [24], and a protein sequence neighbour-joining tree was constructed using the alignments. The parameters included a bootstrap method of phylogeny test, a Poisson model of substitution with 10,000 bootstrap replications and the pairwise deletion of missing data.

## 3. Results

### 3.1. Protein Profiling of Redclaw Tissue by SDS-PAGE

Individual extracts were visualised using SDS-PAGE (Appendix A). Tissue extracts were pooled (*n* = 9 or 10 individuals) to ensure an average representative protein profile for each tissue and treatment. A further SDS-PAGE was performed to characterise the protein profile of raw and heated pooled tissues (Figure 1A,B). Raw claw displayed two major protein bands at 75 and 37 kDa. Heating resulted in a reduction of the 75 kDa band and an increase of the 37 kDa band. Raw tail displayed many protein bands ranging from over 250 kDa to 15 kDa. Heating reduced/removed most bands, with intense bands remaining in the 37 and 15 kDa range. Raw cephalothorax contained many faint bands, the most intense being at 75 kDa. Heating reduced/removed most bands, except for faint bands at 75, 37, 17 and less than 10 kDa.

### 3.2. Identification of Allergens Using Specific Antibodies

Immunoblotting using monoclonal anti-SCP antibody displayed specific SCP binding in all tissues at 23 kDa (Figure 2A,B). However, the cephalothorax also contained a band present at 20 kDa. The tail contained the most SCP, while the cephalothorax and claw contained considerably less. Heating resulted in the SCP-binding bands remaining relatively stable at 23 kDa and 20 kDa, as well as fragmented SCP bands at 50 kDa and 75 kDa in the heated samples.

Immunoblotting using anti-HC polyclonal antibodies was less specific (Figure 2C,D). In raw extracts, large amounts of HC were present, with intense bands in the region between 100 and 75 kDa. Heating reduced the HC-binding bands; however, it varied depending on the type of tissue. Raw claw contained many HC-binding bands present in the 75 kDa region. Upon heating, intense bands remained present at approximately 70 and 60 kDa. The heated tail tissue displayed a great reduction in HC-binding bands compared to the raw, with two bands found at 75 kDa. Raw cephalothorax tissue displayed a great variety of HC-binding bands from 100 to 20 kDa. Upon heating, HC bands were greatly reduced to two faint bands at approximately 100 kDa.

### 3.3. Allergen Abundance in Different Tissues by LC/MS

The relative abundance of proteins identified by LC/MS, measured in percentages of intensity-based absolute quantification (iBAQ%), were grouped based on whether they were: (a) a characterised allergen, (b) a potential allergen (evidence of allergenicity, but not characterised), or (c) a non-allergen (no evidence of allergenicity) (Figure 3). It is to be noted that technically, LC/MS is only able to analyse soluble proteins in the extracts. Heating results in the denaturing of heat-labile proteins, and these proteins become insoluble and are precipitated upon centrifugation. Furthermore, subsequent immunological analysis using immunoblot is not possible with insoluble protein. Using a heated extract in this study may contain some differences from that of heated tissue, as the tissue also contains insoluble proteins. However, comparing the protein composition between raw and heated extracts allows for a good comparison within the treatment groups (raw and heated) and between different tissues.

The proteomic composition of soluble protein varied among the tissues and treatments (Figure 3). Allergens and PAs made up at least 45% of all relative protein abundance. Heating of the claw and tail extracts resulted in an increase in the allergen proportion (Figure 3). However, this change was unexpectedly not found in the cephalothorax, which showed little change in composition after heating.

The protein and peptide counts indicated differences between tissues and treatments (Table 2). Raw claw contained the most unique proteins, followed by the tail and cephalothorax. Despite this, the tail had the most peptides, followed by the claw and cephalothorax. The heated cephalothorax contained the most proteins, followed by the tail and claw. The peptide counts were similar between tissues; the claw contained the most, followed by the tail and then the cephalothorax.

The relative abundance (iBAQ%) data indicated allergen proportion differences between tissues (Figure 4). In the raw claw tissue (Figure 4A), HC, PA and SCP were most abundant. In the raw tail (Figure 4B), SCP, PA, HC, and TM were most abundant. Finally, in the raw cephalothorax (Figure 4C), HC, SCP, and fatty acid-binding protein (FABP) were most abundant. TM was present in small amounts in raw tissues, with the most found in the tail and the least in the cephalothorax. HC was abundant in raw claw and the cephalothorax but more scarce in the tail. SCP was abundant in the raw tail and least in the claw. MLC was present in small amounts, mainly in the raw tail and with the least being in the cephalothorax. FABP was particularly abundant in the cephalothorax but scarce in the claw and tail.

### 3.4. Impact of Heating on Individual IgE Reactivity

Heating resulted in the MLC, TM and troponin becoming abundant in claw and tail tissues (Figure 4A,B). In the cephalothorax tissue (Figure 4C), heating resulted in the MLC, TM and SCP being most abundant. Overall, upon heating, TM greatly increased in proportion (increased by 11% in the claw, 15% in the tail, and 10% in the cephalothorax). However, the heated tail still contained the most TM, and the cephalothorax contained the least. HC was reduced upon heating in all tissues, with the cephalothorax now containing the most HC and the tail the least. SCP was also reduced when heated, though it remained most abundant in the tail and least in the claw. The MLC proportion increased in all tissues and was most abundant in the tail and least in the cephalothorax. However, MLC was consistently abundant in all tissues. Troponin was increased upon heating, the most being in the claw and tail, and the least in the cephalothorax. Finally, FABP was largely reduced upon heating, with the most remaining being in the cephalothorax and the least in the claw.

Sera from 11 adults, 13 paediatrics and two controls were incubated against raw and heated tails to determine IgE-binding (Figure 5). In the raw immunoblots, all adult sera displayed binding to a 75 kDa band, and 72% (*n* = 8) displayed binding to a 58 kDa protein. Another 36% (*n* = 4) of adults displayed binding to a 37 kDa band, and one bound to a 19 kDa band (Figure 5A). In paediatrics, 92% (*n* = 12) displayed binding to bands ranging from 37 to 15 kDa, with only one individual binding to a 21 kDa band (Figure 5B).

Immunoblot using heated tails demonstrated intense binding in adults and paediatrics (Figure 5C,D). In adults, 77% (*n* = 9) of sera bound to proteins in the 40 to 15 kDa region. However, 18% (*n* = 2) of adults still bound to the 75 kDa protein that was found in raw extracts. Binding was similar in paediatrics, 97% (*n* = 12) of sera bound at the 40 to 15 kDa range. However, one paediatric serum only bound to a 21 kDa protein, which was also seen in the heated and raw extracts. Heating resulted in similar binding between adults and paediatrics, with binding regions from 40 to 15 kDa.

### 3.5. Comparative Allergen Analysis of Different Tissues Using Pooled Patient Sera

The immunoblotting of pooled sera resulted in distinct IgE-binding bands in all extracts (Figure 6). Raw claw contained two distinct bands, one at 75 kDa and another at 37 kDa. Heating increased the intensity of the 37 kDa band and two faint bands at 20 kDa. In raw tails, one distinct band was present at 37 kDa. Heating resulted in IgE-binding bands at 100, 75, 37, 25 and 20 kDa, the most intense of which was 37 kDa. In raw cephalothorax, protein bands binding to IgE antibodies were found at 37, 19 and 17 kDa. Once heated, the bands observed at 19 and 17 kDa in raw cephalothorax were absent.

### 3.6. Quantitative Allergen Analysis of IgE-Reactive Proteins by LC/MS

IgE-binding bands from pooled sera were excised from the SDS-PAGE gel (Appendix A) and analysed by liquid chromatography-mass spectrometry (LC/MS) to identify the protein’s identity and relative abundance. The complete protein content of each analysed band is presented in the (Appendix A). Of the four most abundant proteins in each band (see Appendix A), the top allergen per band is presented below (Table 3, Table 4, Table 5, Table 6, Table 7 and Table 8).

### 3.7. Amino Acid Sequence Analysis of Allergens in Redclaw and Various Crustacean Species

Amino acid sequences of the predominantly found allergens TM, AK, HC, MLC, SCP, and troponin C were compared between redclaw and various crustacean species (*Cherax destructor, Homarus americanus, Homarus gammarus, Litopenaeus vannamei, Penaeus monodon, Procambarus clarkii* and *Scylla paramamosain*) (Figure 7). NBCI accession numbers for all amino acid sequences are included in Figure 7 In-depth phylogenetic analysis of the analysed crustacean species and each species-specific allergen can be found in the (Appendix A).

Redclaw TM displayed a high sequence similarity between species and was 100% identical to the two other crayfish and was least similar to both crabs (~93% similar) (Figure 7A). Redclaw AK also displayed a high sequence similarity and was most similar to the two other crayfish (98.60% and 96.6% similar, respectively) and was most different from both shrimps (90.1% and 90.7% similar, respectively) (Figure 7B). Redclaw HC was most similar to the shrimp *L. vannamei* (74.2% similar)*,* and least to the crab *S. paramamosain* (56.4% similar) (Figure 7C). Redclaw MLC was highly identical to both shrimp MLCs (90.4% similar to both), while it was least similar to crayfish *P. clarkii* MLC (18.8% similar), and the MLC from both crabs *S. paramamosain* (19.6% similar) and *P. trituberculatus* (19.4% similar) (Figure 7D). Redclaw SCP was most similar to the SCP from both shrimp species (74.2% and 73.2% similarity, respectively) and most different from the crab species *P. trituberculatus* (56.4% similar) (Figure 7E). The last allergen investigated was troponin C, in which all crustacean sequences were about 50% similar to the redclaw troponin C (Figure 7F).

## 4. Discussion

The allergenic differences among various shellfish tissues remain under-investigated. This lack of knowledge relating to tissue-specific allergens may result in missing PAs and hinder the effective diagnosis of a shellfish allergy in patients. In our study, tissue-specific differences in allergens were examined among three body parts and indicated that extracts from the redclaw claw and tail are most similar in allergen content compared to the extracts from the cephalothorax. Key allergens in the claw/tail included HC, MLC, SCP, TM and troponin, as well as other PAs that have evidence of IgE-binding in patients but have not been registered in the WHO/IUIS. In contrast, primary allergens in cephalothorax tissues included FABP, HC, SCP and MLC. Additionally, it was found that the relative abundance (iBAQ%) of predominant allergens varies amongst body parts. Finally, the results from SDS-PAGE, immunoblot and LC/MS indicate that previously described heat-stable allergens are only partially heat-stable, and allergenicity is reduced upon heating. For example, studies into HC regard it as being heat-stable [25,26], whereas others suggest that allergenicity is reduced upon heating [27]. However, reductions are different depending on the tissue type and allergen. The results suggest evidence for allergen isoforms with varying heat stability in the different tissues, although there are limited studies that have investigated the presence of allergen isoforms in shellfish. In those conducted, HC appears to exhibit many isoforms, with 12 isoforms found in tiger shrimp *P. monodon* [11]. HC isoforms display unique sequence diversity and distinctive tissue expression patterns. As a result, this study may suggest that allergen isoforms also have varying allergenicity. However, the differences between allergen isoforms have not been investigated in the context of altered allergenicity. The findings of the previous study are supported by isoform expression studies in crabs (*Cancer magister*) and shrimp (*P. monodon)* [28,29]. Different isoforms were expressed under different conditions (e.g., cold stress, hypoxia, etc.) [28]. In shellfish, the differences between isoform allergenicity have not been investigated and are required for improved component-resolved diagnosis.

Although studies determining allergenicity between shellfish tissues are limited, the initial evidence of tissue-based differences supports the findings from this current study. It was reported that shellfish-sensitive patients may test negative to conventional IgE tests using shellfish tail muscle extract but have allergic symptoms upon consuming the cephalothorax [10]. However, when performing a prick-to-prick test using cephalothorax extract instead, patients tested positive [10]. A subsequent study found that HC in shrimp cephalothorax triggered anaphylaxis in some individuals. Patients with a history of anaphylaxis to shrimp but negative IgE test results underwent prick-to-prick tests using *L. vannamei* tail and cephalothorax. The patients tested negative for the tail but positive for the cephalothorax [9]. Immunoblot analysis using patient sera found two IgE-binding bands at 75 kDa that were identified as HC [9]. These studies implicate HC as a major allergen in cephalothorax due to its high abundance and potential to cause patient anaphylaxis [9]. The presence of this HC was confirmed in redclaw, which is taxonomically in a different group to crabs and shrimps.

Determining the presence of allergens in raw and heated shellfish is important for consumption and food processing. Heat processing removes many heat-labile proteins but increases the proportion of heat-stable proteins. As detailed in Section 3.3, centrifugation results in the precipitation of insoluble, denatured proteins and allergens in heated extracts. This results in some differences between the heated tissue and the heated extract, as the extract no longer contains insoluble protein, while it remains in heated tissue. However, a standardised treatment of protein extracts allows for a comparison between raw and heated extracts derived from claw, tail and cephalothorax. Heating typically increases the patient’s recognition of stable allergens and enhances IgE-binding [9]. In redclaw, TM and MLC were present in raw tissue at relatively low amounts but, upon heating, became predominantly abundant allergens. Antibody detection of TM has previously been shown to increase upon heating in many shellfish species, which may also be due to conformational changes in the secondary structure [11,30].

TM, as the most reported major shellfish allergen, remains focal in heat-processing studies [12], with most other allergenic proteins often overlooked. In this present study, TM, troponin and MLC, proportionally against other proteins, increased when heated, while HC, SCP and PA decreased. However, HC has been described in previous studies to be heat-stable [25]. In redclaw, the proportion of HC was reduced in all extracts upon heating. However, HC still demonstrated antibody binding upon heating. Similar to HC, SCP is considered to be heat-stable [31,32]. However, our results indicated that SCP binding decreases upon heating but is not completely removed. Instead, bands of SCP are found at high molecular weights, which are likely to be SCP aggregates of degraded fragments. SCP aggregates also demonstrate antibody-binding epitopes to the shrimp-specific anti-SCP antibody. This may indicate that perhaps SCP allergenicity decreases upon heating. Other PAs (allergens in similar species) have not been characterised in shellfish; thus, their heat stability is unknown. Research into characterising PAs and determining their allergenicity is required. The current study found that heat stability involves certain allergen isotypes, and some allergens are only partially heat-stable. In our study, redclaw of the same age were sourced from the same hatchery and thereby are standardised for possible protein variations due to age or environmental conditions. Other studies indicate that allergenic protein isoforms may be upregulated or downregulated depending on factors such as age or the moult cycle [28,29]. As a result, our current data indicates that in age-matched specimens, there is potential for varying heat stability between allergen isoforms, and many allergens are not strictly heat-stable or labile as previously assumed.

A comparison of amino acid sequences sourced from the protein NCBI database indicates that redclaw TM, AK, HC and SCP are similar (some even identical) to other crustaceans. In contrast, MLC varies depending on the species that redclaw is compared against. For example, redclaw MLC was similar to penaeid shrimp but divergent to crab MLC and even to that of the crayfish *Procambarus clarkii* (Figure 7). However, more research into the similarities between the IgE-binding epitopes of redclaw and other crustacean species is needed to determine if these sequence differences are likely to affect allergenicity. IgE-binding epitopes may be highly conserved regions in the amino acid sequence with little to no sequence changes between species. Specifically, TM, MLC and SCP are all highly cross-reactive proteins between most shellfish and potentially some insect species [33,34,35,36,37]. TM is a major invertebrate pan-allergen due to containing high sequence similarities among crustaceans, molluscs, arachnids and insects [37]. Thus, patients sensitive to TM are likely sensitive to many arthropod species. In contrast, MLC and SCP have been scarcely investigated compared to TM. Shrimp MLC contains high sequence similarities to insects such as the cockroach [36]. Meanwhile, SCP sequences are similar between shrimp and crayfish species but considerably different from molluscs [34,38]. Previous SCP sequence studies indicate that SCP may not be as highly conserved as MLC and TM, which supports the present results, as SCP has a lower sequence similarity between redclaw and shrimp species.

Our study demonstrates that adult and paediatric patients were sensitised to various proteins in different redclaw tissues. MLC and SCP from *L. vannamei* were found to be of increased importance in paediatrics, as the frequency of recognition of these allergens is over double that of adults (SCP: 70% vs. 31% in adults, and MLC: 59% vs. 21% in adults). Paediatric patients have also demonstrated increased epitope recognition of many known shrimp allergens. The frequency of allergen recognition is greater in paediatrics, with recognition being almost twice that of adults [39].

When exposing redclaw extracts to shrimp-sensitive sera, TM was found to be a common IgE-binding protein in both adults and paediatrics. This is reflected in the literature, as adult and paediatric populations were shown to have high recognition frequencies, with up to 61% and 94% to TM from *L. vannamei* [39]. This suggests that TM is associated with the persistence of shellfish allergy into adulthood [36,39]. Interestingly, in this present study, adult sera recognised HC (75 kDa) in raw extracts more frequently than in paediatrics. However, three paediatric sera displayed binding to HC. Increased binding in adults may signify that sensitivity and recognition of HC become more prevalent with age. Studies have also found that HC is likely cross-reactive with shellfish and insects, including house dust mites (HDM) and cockroaches. In a study by Kamath et al., 2017 [40], over 60% of HDM-sensitive infants reacted to HC from Turkish narrow-clawed crayfish (*Astacus leptodactylus*). This could indicate that crayfish HC shares cross-reactive epitopes with HDM HC [40]. In tropical regions (e.g., northern Australia and Southeast Asia), allergic sensitisation is often presented to the tropical mite (*Blomia tropicalis)* [41]. However, *B. tropicalis* allergens are still not well characterised.

Serum IgE-binding patterns and identified allergens vary among body parts. SDS-PAGE and immunoblot, using pooled sera, display similar binding patterns between claw and tail extracts, with a large region at 37 kDa present in all extracts. The 37 kDa band, identified as TM, was found primarily in the claw and tail, as compared to the cephalothorax. These results support previous findings, as TM is considered a major protein involved in muscle contraction [37]. Therefore, TM would be more present in the claw and tail, as these body parts comprise primarily fast and slow-contracting muscles. The results from the immunoblot and LC/MS indicate that HC is more abundant in the claw and cephalothorax, with similar relative abundance levels in the 75 kDa bands. These results are supported by the literature, in which HC was found to be synthesised in the hepatopancreas; thus, higher levels of certain HC isoforms were predominantly found in cephalothorax tissue [42]. The results for claw also indicate a greater presence of IgE-binding to HC. Despite this, there is little information regarding allergenicity and the presence of HC in claw tissue.

The results from the pooled serum immunoblot and LC/MS analysis identified the main IgE-binding proteins. The most allergenic isoform of TM is TM isoform X17, predominantly found at 37 kDa in heated extracts, as well as in lower abundances at 37 kDa in raw extracts. This suggests TM isoform X17 is a protein found in all muscle tissues and seems to be the predominant allergenic form of TM in redclaw. Differences in predominant TM sequences between redclaw, as well as other various crayfish and shrimp species, have been documented [14]. Redclaw TM IgE-binding epitopes were found to contain no alteration in the conserved regions compared to TM from *Scylla serrata, L. vannamei*, *Exopalaemon modestus*, *Macrobrachium rosenbergii*, *Macrobrachium lanchesteri, and P. clarkii* [14]. This indicates that patients sensitive to other crustaceans are likely sensitive to redclaw, as TM is extremely conserved amongst shellfish [14].

The LC/MS results indicate that in heated tails, MHC is the most abundant protein in the IgE-binding band. However, MHC is not a registered WHO/IUIS allergen despite being part of the same complex as MLC. Recent studies indicate MHC allergenicity in patients with fish allergy [43] and increasing evidence in shellfish allergy. MHC is a potential allergen in *P. monodon* [44,45] and is cross-reactive with HDM MHC [44]. Studies on banana shrimp (*Fenneropenaeus merguiensis*) have also found evidence of MHC allergenicity in shrimp-sensitive patients [46]. The results from this study may also indicate the allergenicity of MHC. It is worth noting that despite this, this binding may come from the second most abundant protein present in these bands, TM isoform X17. As shown above, TM isoform X17 seems to be a major allergen isoform that causes intense IgE-binding in shrimp-allergic patients. However, inhibition studies are required to determine the protein responsible for this binding.

## 5. Conclusions

In conclusion, differences in allergen content were established between different types of redclaw tissues and the allergenicity of redclaw proteins was characterised against adult and paediatric shrimp-allergic patients. Between the different body tissues, the claw and tail have a similar proteomic composition in contrast to the cephalothorax, which was consistently different. Heat treatment significantly alters the proportion of soluble allergenic proteins in redclaw extracts; however, this was different depending on the tissue. Importantly, allergens TM, AK, HC and SCP were highly conserved between redclaw and commonly consumed crustaceans (*C. destructor*, *P. clarkii*, *L. vannamei*, *P. monodon*, *S. paramamosain*, *P. trituberculatus*, *H. gammarus and H. americanus*), whereas MLC was only highly conserved between redclaw and shrimp species (*L. vannamei* and *P. monodon)*. This protein similarity was reflected in the IgE cross-reactivity with shrimp-allergic patients. The findings of this study will provide fundamental knowledge for improved diagnostics in shellfish allergy with a focus on differences in allergenicity of different body tissues. However, treatment of shellfish allergy, including redclaw allergy, remains limited to avoidance of the implicated source. The findings of this study may also be used to improve shellfish-allergy treatment options, including improvement in immunotherapeutic approaches and diagnosis.

## Figures and Tables

**Figure 1 foods-13-00315-f001:**
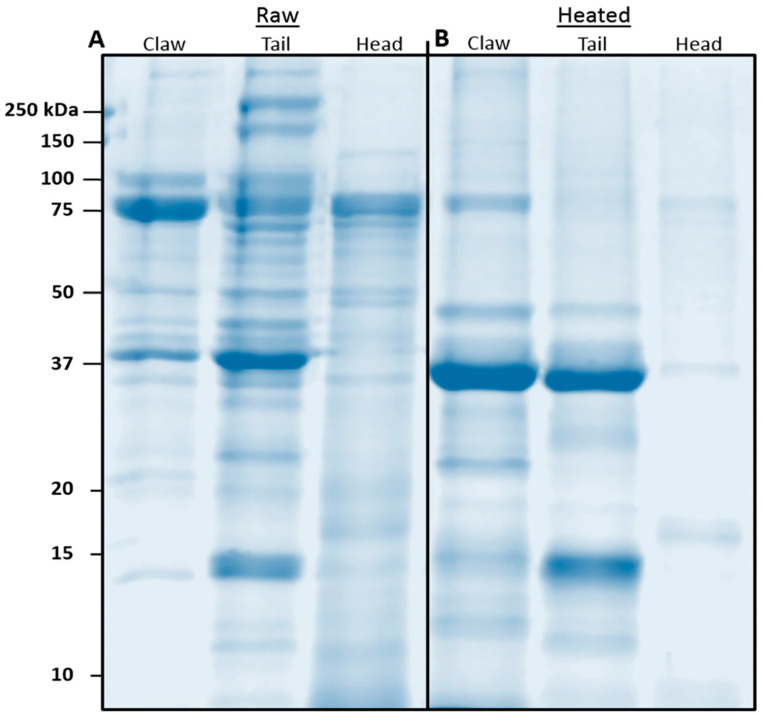
Reducing SDS-PAGE of pooled (**A**) raw and (**B**) heated redclaw tissue extracts from claw, tail and cephalothorax (CPTX). Samples containing 5 μg of protein (representing approximately 147–153 ug of tissue) were run on AnyKD SDS-acrylamide gels at 170 V for 1 h.

**Figure 2 foods-13-00315-f002:**
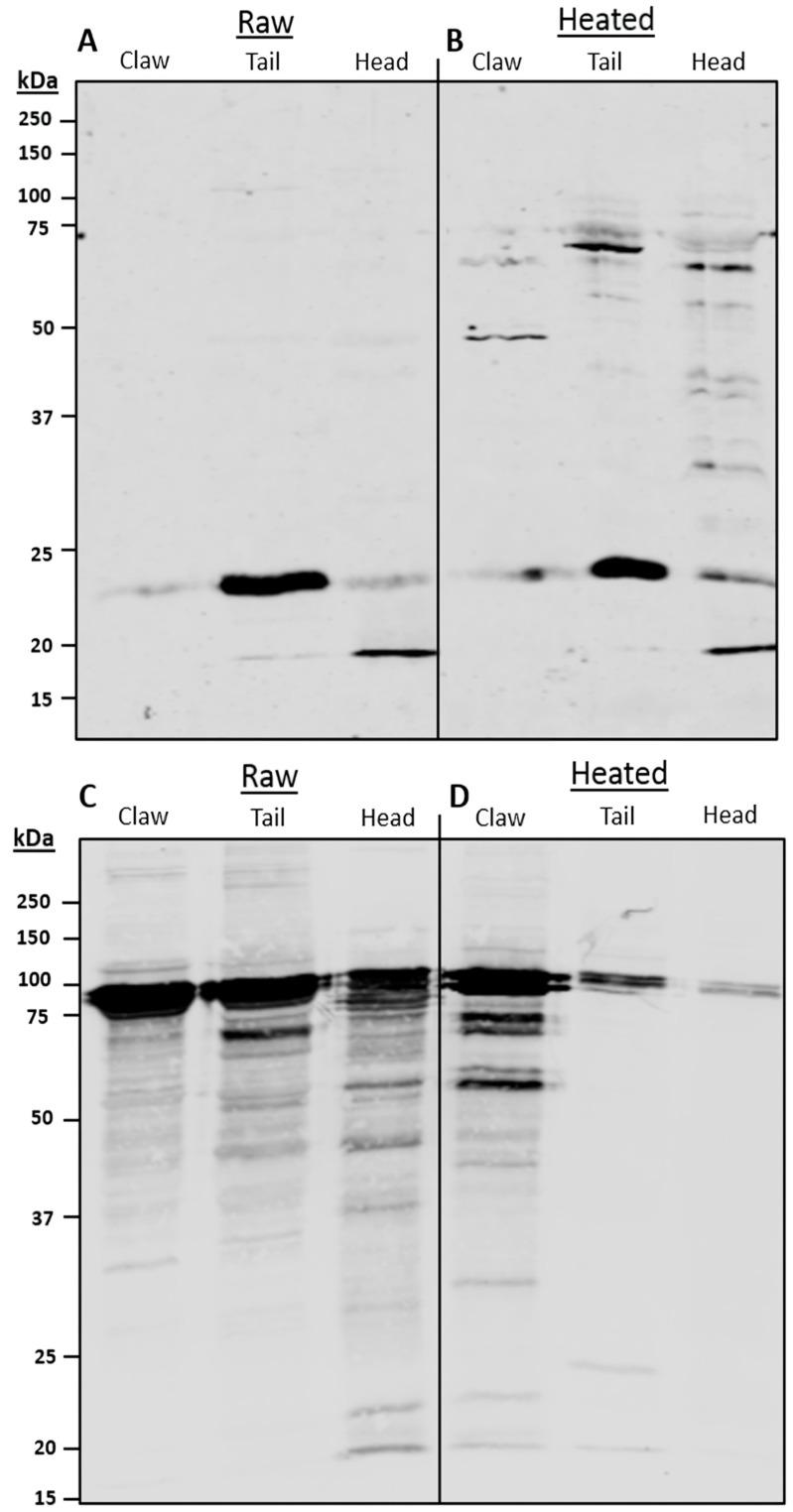
Antibody immunoblotting analysis of redclaw tissue extracts. Using (**A**,**B**) monoclonal anti-SCP antibody and (**C**,**D**) polyclonal anti-HC antibody for raw (**A**,**C**) and heated (**B**,**D**) redclaw extracts from the claw, tail and cephalothorax (CPTX).

**Figure 3 foods-13-00315-f003:**
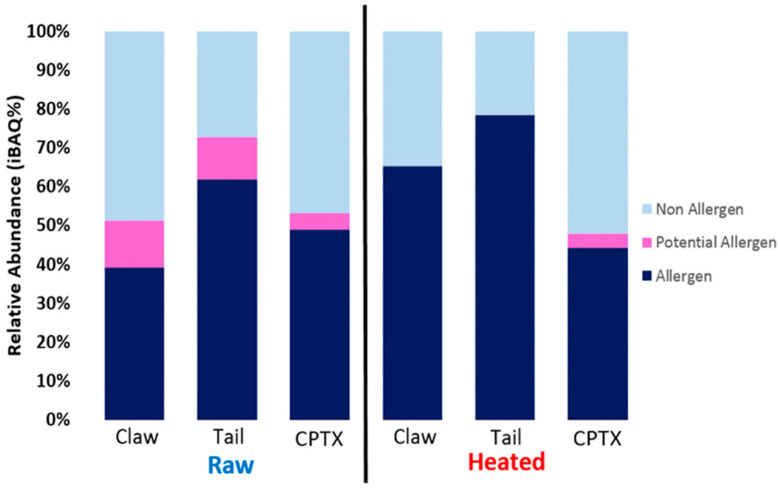
Composition of redclaw tissue extract proteomes. LC/MS results were searched for 11 major shellfish allergens and five PAs. The relative abundance values (iBAQ%) for all major allergens, PAs (uncharacterised proteins that have evidence of allergenicity) and non-allergens (other proteins) were totalled and grouped.

**Figure 4 foods-13-00315-f004:**
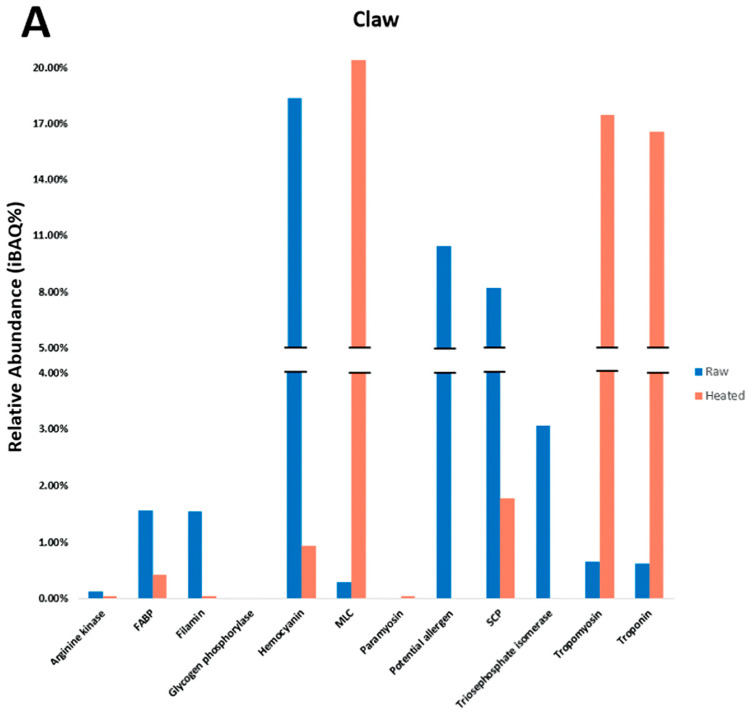
Relative abundance (iBAQ%) results for raw and heated redclaw tissues. Relative abundance indicates the proportion of a certain protein in a sample as found using LC/MS. Major allergenic proteins were included (*n* = 11), as well as six PAs that have evidence of allergenicity. Samples include raw and heated (**A**) claw, (**B**) tail and (**C**) cephalothorax (CPTX).

**Figure 5 foods-13-00315-f005:**
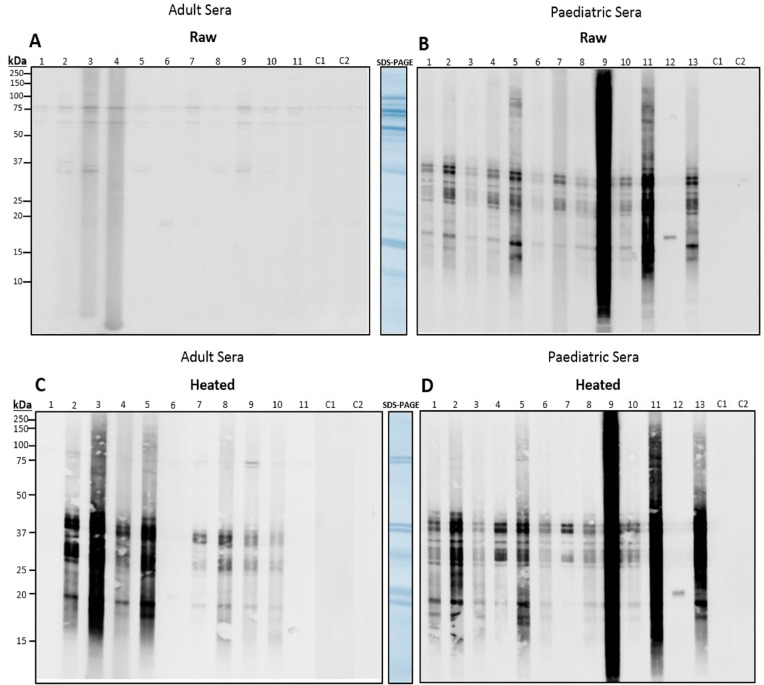
Individual adult and paediatric immunoblots against raw and heated redclaw tail extract. Patients included 11 adults (**A**,**C**) and 13 paediatrics (**B**,**D**) sensitive to shellfish mix, blotted against raw (**A**,**B**) and heated (**C**,**D**) pooled redclaw tails. Two patients with no demonstrated sensitivity to shellfish were used as negative controls.

**Figure 6 foods-13-00315-f006:**
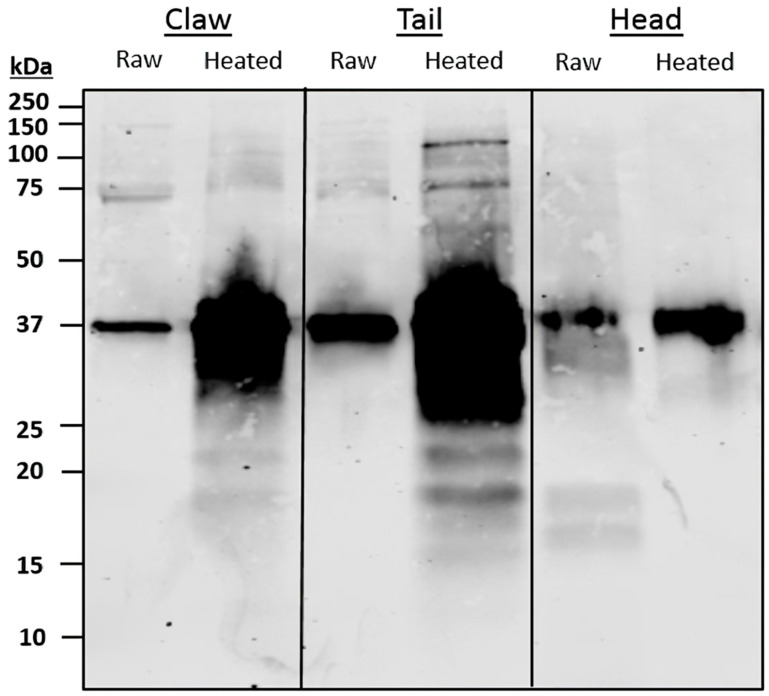
Pooled patient sera immunoblot against pooled redclaw claw, tail and cephalothorax (CPTX) samples. Three adult and paediatric sera were pooled and incubated on a membrane against pooled redclaw samples to determine IgE-binding bands.

**Figure 7 foods-13-00315-f007:**
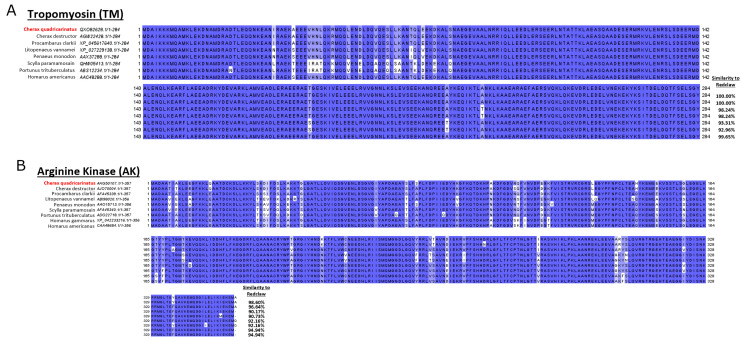
Sequence alignments of (**A**) TM, (**B**) AK, (**C**) HC, (**D**) MLC, (**E**) SCP and (**F**) troponin C between redclaw and various crustacean species. Sequence alignments demonstrate percentage identity between redclaw (*Cherax quadricarinatus*) TM, AK, HC, MLC, SCP and troponin C against various crustacean species (*Cherax destructor*, *Procambarus clarkii*, *Litopenaeus vannamei*, *Penaeus monodon*, *Homarus americanus*, *Homarus gammarus*, *Portunus trituberculatus* and *Scylla paramamosain*).

**Table 1 foods-13-00315-t001:** Allergenic and potentially allergenic proteins were searched for using MaxQuant v2.2. Searched proteins included 11 known allergens and five potential allergens (PAs) found in *Cherax quadricarinatus*, as seen by the protein accession number and molecular weight (MW) of protein.

**Accession Number**	**Allergen**	**MW of Protein**
AKG50107.1	Arginine kinase	50 kDa
AGB13925.1	Fatty acid-binding protein	14 kDa
XP_053627230.1	Filamin	261 kDa
XP_053641175.1	Glycogen phosphorylase-like protein	97 kDa
AFP23115.1	Hemocyanin	78 kDa
XP_053648994.1	Myosin light chain	20 kDa
XP_053642016.1	Paramyosin	102 kDa
XP_053635744.1	Sarcoplasmic calcium-binding protein	21 kDa
XP_053651606.1	Triosephophosphate isomerase	29 kDa
QXO82628.1	Tropomyosin	32 kDa
XP_053653993.1	Troponin	17 kDa
**Accession Number**	**Potential Allergen**	**MW of Protein**
XP_053656740.1	Enolase	47 kDa
XP_053639126.1	Fructose-1,6-bisphosphatase	36 kDa
XP_053629118.1	Pyruvate kinase	59 kDa
XP_053651900.1	Titin (fragment)	N/A (depends on fragment)
AAG17936.1	Vitellogenin	291 kDa

**Table 2 foods-13-00315-t002:** Protein and peptide counts for all redclaw tissue extracts as determined using liquid chromatography-mass spectrometry (LS/MS) and MaxQuant analysis against the NCBI database of the suborder Pleocyemata. (Cephalothorax is abbreviated to CPTX).

	Raw			Heated		
Redclaw	Claw	Tail	CPTX	Claw	Tail	CPTX
**Proteins (N=)**	1357	1273	898	631	1079	1193
**Peptides (N=)**	11,949	12,421	4990	10,638	10,068	9513

**Table 3 foods-13-00315-t003:** Most abundant IgE-binding proteins in raw claw analysed using liquid chromatography-mass spectrometry. Most abundant proteins found in bands 1 to 4, including band molecular weight (kDa), protein ID (NCBI), relative abundance (iBAQ%), coverage (%) and protein molecular weight (kDa).

Band	MW of Band	Protein ID (NCBI)	Most Abundant Protein	Relative Abundance (iBAQ%)	Coverage (%)	MW of Protein
1	75 kDa	AFP23115.1	Hemocyanin	61.66%	31.5%	78 kDa
2	74 kDa	XP_053627537.1	Hemocyanin B chain-like	38.44%	48.8%	78 kDa
3	37 kDa	XP_053645744.1	Arginine kinase	78.21%	42.2%	50 kDa
4	36 kDa	XP_053645744.1	Arginine kinase	70.25%	36.2%	50 kDa

**Table 4 foods-13-00315-t004:** Most abundant IgE-binding proteins in heated claw analysed using liquid chromatography-mass spectrometry. Most abundant proteins found in bands 1 to 4, including band molecular weight (kDa), protein ID (NCBI), relative abundance (iBAQ%), coverage (%) and protein molecular weight (kDa).

Band	MW of Band	Protein ID (NCBI)	Most Abundant Protein	Relative Abundance (iBAQ%)	Coverage (%)	MW of Protein
1	75 kDa	AFP23115.1	Hemocyanin	29.25%	38.4%	78 kDa
2	37 kDa	XP_053648131.1	Tropomyosin isoform X17	94.6%	81%	33 kDa
3	20 kDa	XP_053629360.1	Myosin light chain 2	50.82%	13.8%	19 kDa
4	17 kDa	XP_053649409.1	Troponin C isoform X2	56.36%	59.4%	17 kDa

**Table 5 foods-13-00315-t005:** Most abundant IgE-binding proteins in raw tail analysed using liquid chromatography-mass spectrometry. Most abundant proteins found in bands 1 to 3, including band molecular weight (kDa), protein ID (NCBI), relative abundance (iBAQ%), coverage (%) and protein molecular weight (kDa).

Band	MW of Band	Protein ID (NCBI)	Most Abundant Protein	Relative Abundance (iBAQ%)	Coverage (%)	MW of Protein
1	75 kDa	AFP23115.1	Hemocyanin	20.42%	42.6%	78 kDa
2	37 kDa	XP_053645744.1	Arginine kinase	70.64%	53.7%	33 kDa
3	37 kDa	XP_053645744.1	Arginine kinase	72.89%	49.7%	50 kDa

**Table 6 foods-13-00315-t006:** Most abundant IgE-binding proteins in heated tail analysed using liquid chromatography-mass spectrometry. Most abundant proteins found in bands 1 to 12, including band molecular weight (kDa), protein ID (NCBI), relative abundance (iBAQ%), coverage (%) and protein molecular weight (kDa).

Band	MW of Band	Protein ID (NCBI)	Most Abundant Protein	Relative Abundance (iBAQ%)	Coverage (%)	MW of Protein
1	100 kDa	XP_053645272.1	Myosin heavy chain	29.58%	34.8%	218 kDa
2	75 kDa	XP_053645272.1	Myosin heavy chain	27.34%	34.2%	218 kDa
3	57 kDa	XP_053648131.1	Tropomyosin isoform X17	41.06%	54.9%	33 kDa
4	50 kDa	XP_053648131.1	Tropomyosin isoform X17	95.28%	81.7%	33 kDa
5	38 kDa	XP_053648131.1	Tropomyosin isoform X17	93.17%	83.5%	33 kDa
6	37 kDa	XP_053648131.1	Tropomyosin isoform X17	98.14%	84.5%	33 kDa
7	31 kDa	XP_053648131.1	Tropomyosin isoform X17	71.36%	69.4%	33 kDa
8	25 kDa	XP_053648131.1	Tropomyosin isoform X17	28.13%	64.4%	33 kDa
9	29 kDa	XP_053629360.1	Myosin light chain 2	78.87%	89.1%	19 kDa
10	18 kDa	XP_053629360.1	Myosin light chain 2	50.33%	62.1%	19 kDa
11	17 kDa	XP_053656247.1	Myosin light chain 1 isoform X3	45.10%	57.9%	15 kDa
12	16 kDa	XP_053656247.1	Myosin light chain 1 isoform X3	27.86%	57.9%	15 kDa

**Table 7 foods-13-00315-t007:** Most abundant IgE-binding proteins in raw cephalothorax analysed using liquid chromatography-mass spectrometry. Most abundant proteins found in bands 1 to 5, including band molecular weight (kDa), protein ID (NCBI), relative abundance (iBAQ%), coverage (%) and protein molecular weight (kDa).

Band	MW of Band	Protein ID (NCBI)	Most Abundant Protein	Relative Abundance (iBAQ%)	Coverage (%)	MW of Protein
1	75 kDa	AFP23115.1	Hemocyanin	29.67%	38.7%	78 kDa
2	57 kDa	XP_053627537.1	Hemocyanin B chain	19.59%	39.3%	78 kDa
3	37 kDa	XP_053648131.1	Tropomyosin isoform X17	46.47%	62.3%	33 kDa
4	19 kDa	XP_053634537.1	Sarcoplasmic calcium-binding protein isoform X3	46.73%	62.7%	22 kDa
5	17 kDa	XP_053635594.1	Sodium/calcium exchanger regulatory protein 1	9.46%	57.4%	16 kDa

**Table 8 foods-13-00315-t008:** Most abundant IgE-binding proteins in heated cephalothorax analysed using liquid chromatography-mass spectrometry. Most abundant proteins found in bands 1 and 2, including band molecular weight (kDa), protein ID (NCBI), relative abundance (iBAQ%), coverage (%) and protein molecular weight (kDa).

Band	MW of Band	Protein ID (NCBI)	Most Abundant Protein	Relative Abundance (iBAQ%)	Coverage (%)	MW of Protein
1	75 kDa	AFP23115.1	Hemocyanin	29.28%	34.1%	78 kDa
2	37 kDa	XP_053648131.1	Tropomyosin isoform X17	94.25%	70.1%	33 kDa

## Data Availability

Data is contained within the article or Appendix A.

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
