# Peer review of "Allergen Diversity and Abundance in Different Tissues of the Redclaw Crayfish (Cherax quadricarinatus)"

_foods, 2024, doi:10.3390/foods13020315_

Round 1

Reviewer 1 Report

Comments and Suggestions for Authors

The authors present a paper on “The allergen diversity and abundance in different tissues of the redclaw crayfish (Cherax quadricarinatus) by Emily M. Jerry and colleagues. They characterized allergenic proteins in various red claw body tissues (tail, claw, and cephalothorax) and determined how the stability of allergenic proteins was affected by cooking (raw vs. cooked tissues). The manuscript needs minor improvement; however, the authors are asked to take the following recommendations into account to improve their manuscript:

  1. It is not mentioned in the methodology nor in the Figures´ captions which molecular weight marker was used in SDS gels.

2.     Fig 1. also shows a protein band of less than 10 kDa after heating. Have they found an allergen of that molecular weight or lower?

3.     Human Research Ethics Committee Approval Number: H4313 and H6829. This approval number does not indicate the institution from which they obtained them.  Was the patient's consent obtained?

4.     Figure on Page 10, probably Fig. 4, has no caption, and the letters are tiny and unreadable. This figure could be sent to Supplementary material.

5.     The letters in Figure 7 are tiny and unreadable.

6.      The results in Section 3.6, “Quantitative Allergen Analysis of IgE-reactive Proteins by LC/MS, " could be included in the tables; the paragraph is too long and hard to read. One part of the Table is for the proteins in raw material, and the other is for proteins in heated tissues.

7.     In the conclusions section, the minimum composition of redclaw allergens in a skin prick test, for instance, could be stated or suggested.

Comments on the Quality of English Language

The last parragraph in page 12 could be substituted by  Tables 3-8.

Reviewer 2 Report

Comments and Suggestions for Authors

The manuscript “Allergen Diversity And Abundance In Different Tissues Of The Redclaw Crayfish (Cherax quadricari-natus)” describes an investigation of the allergen pattern in three different tissues of redclaw crayfish. This study is very interesting and the results can contribute to improve diagnosis and therapy of shellfish allergy. The used methodologies are appropriate, and the language and presentation are fine.

However, I am not convinced about the interpretation of some results, in particular the results from heating experiments. I think that these experiments can give indications about the thermal stability of the proteins/allergens contained in the extracts, but they cannot give information about the concentration of the allergens in the different tissues of redclaw crayfish. Therefore, in my opinion, it is important to clearly state that the changes in the protein/allergen profiles observed after heating are related to the extracts, rather than to the tissues. The cooked crayfish tissues could have allergen/protein profiles with some differences compared to those observed in  the heated extracts. Therefore, I would suggest to check the entire text and change the parts that can cause confusion.

Some specific comments.

1. Abstract. “The potential of redclaw allergens to cross-react with shellfish-allergic patients was also investigated.” This sentence is not clear to me, perhaps it could be restructured, because it is strange that allergens cross-react with patients.

2. Materials and Methods section, paragraph 2.2. “…boiled at 90°C…”

Why did water boilled at 90°C? Was the pressure lower than 1 atmosphere? was the temperature measured in the extract solution? Should the word "boiled" be changed to something like "incubated or kept"? 

3. Materials and Methods section, paragraph 2.2.  “…centrifuged at 20,000 x g for 20 min at 4 °C.”

Why the centrifugation was carried out? To remove the prepcipitated proteins? However, if we cook the crayfish, the proteins remain in the tissue, even those that are denatured. What do we know about the allergenicity of denatured/precipitated proteins? Therefore, the correlation between the allergen profile in the extracts and that in the tissues is limited. In the heated and centrifuged extracts we could have the proteins/allergens still soluble after extraction and heating, rather than all the proteins (including those soluble, those insoluble, those denatured, and so on) present in the tissue.

4. Materials and Methods section, paragraph 2.5. “…monoclonal mouse anti-shrimp sarcoplasmic calcium-binding protein (SCP) (1:1,000 dilution), or polyclonal goat anti-shrimp hemocyanin (HC) (1:50,000 dilution) primary antibodies.” Which was the source of these antibodies? were they available on the market or were they home-produced?

5. Legend to Figure 1. “SDS-PAGE…”

I assume it is a "reducing" SDS-PAGE. However, it is better to add this detail.

6. Legend to Figure 1. “Samples contained 5 µg of protein…”

Is it possible to calculate the amount (for instance in g) of tissue needed to obtain the aliquot loaded on SDS-PAGE (5 ug)?

7. Results, Paragraph 3.2 and elsewhere. “…hemocyanin (HC) …”

This is a repetition, the acronym "HC" has been already reported before. After the first statement, the acronym only should be used in the text.

8. Results, Paragraph 3.2. “…at 100 and 75 kDa…”

Should it be "in the region between 100 and 75 kDa"?

9. Results, Paragraph 3.3. “Heating resulted in MLC, TM and troponin becoming abundant in claw and tail tissues (Figures 4A and B).

We can suppose that all the proteins were present, at the same concentration, in heated and not heated tissues. Nevertheless, we observe differences in the concentration of proteins because  the soluble proteins only can be detected in the extracts. Denatured proteins can become insoluble and precipitate. Therefore, we do not see them in the extract of soluble proteins, because these proteins precipitated, but it does not mean that they are no more present in the heated tissue. The text of the entire manuscript can probably be revised with this concept in mind.

10. I do not see the legend to the Figure that should be Figure 4.

11. Conclusions.  “Heat treatment significantly alters the proteins present in redclaw; however, this was different depending on the tissue.”

Do the Authors mean that "Heat treatment of the extract, followed by centrifugation to remove precipitated proteins, significantly alters the protein pattern present in redclaw extracts"?

Reviewer 3 Report

Comments and Suggestions for Authors

Some figures need high resolution to be more clarified.

Conclusion need that you propose suggestions for this ALLERGEN to be cured.
